# Nuclear α-Synuclein-Derived Cytotoxic Effect via Altered Ribosomal RNA Processing in Primary Mouse Embryonic Fibroblasts

**DOI:** 10.3390/ijms24032132

**Published:** 2023-01-21

**Authors:** Dong Hwan Ho, Hyejung Kim, Daleum Nam, Jinju Heo, Ilhong Son

**Affiliations:** 1Curahora Inc. Project 500 Tower #1502, 311, Anyang-ro, Manan-Gu, Anyang-si 14118, Gyeonggi-do, Republic of Korea; 2InAm Neuroscience Research Center, Sanbon Medical Center, College of Medicine, Wonkwang University, 321, Sanbon-ro, Gunpo-si 15865, Gyeonggi-do, Republic of Korea; 3Department of Neurology, Sanbon Medical Center, College of Medicine, Wonkwang University, 321, Sanbon-ro, Gunpo-si 15865, Gyeonggi-do, Republic of Korea

**Keywords:** Parkinson’s disease, α-synuclein, nucleolin, ribosomal RNA processing

## Abstract

α-Synuclein (αSyn) is an important player in Parkinson’s disease (PD) pathogenesis. The aggregation of αSyn is mainly formed in the cytoplasm, whereas some αSyn accumulation has also been found in the nuclei of neurons. To assess the effect of nuclear αSyn, we generated αSyn conjugated with a nuclear export signal (NES) or a nuclear localization signal (NLS), and compared them with wild-type αSyn in primary mouse embryonic fibroblasts (MEF) using DNA transfection. Overexpression of NLS-αSyn increased cytotoxicity. The levels of apoptotic markers were increased by NLS-αSyn in MEF. Interestingly, an increase in the levels of 40S ribosomal protein 15 was observed in MEF expressing NLS-αSyn. These MEF also showed a higher 28S/18S rRNA ratio. Intriguingly, the expression of NLS-αSyn in MEF enhanced segmentation of nucleolin (NCL)-positive nucleolar structures. We also observed that the downregulation of NCL, using shRNA, promoted a relatively higher 28S/18S rRNA ratio. The reduction in NCL expression accelerated the accumulation of αSyn, and NCL transfection enhanced the degradation of αSyn. These results suggest that nuclear αSyn contributes to the alteration in ribosomal RNA processing via NCL malfunction-mediated nucleolar segmentation, and that NCL is a key factor for the degradation of αSyn.

## 1. Introduction

Parkinson’s disease (PD) is the second most common neurodegenerative disease in the world. The pathological marker of PD is the loss of dopaminergic neurons in the substantia nigra pars compacta (SNpc) [1,2]. Neurodegeneration of dopaminergic neurons is also associated with various types of cell death, including apoptosis, necrosis, and autophagic death [3,4]. However, the mechanism underlying the death pathway in dopaminergic neuronal degeneration is not completely understood. Although the signaling pathway involving molecules responsible for degeneration is partially veiled, the accumulation of α-synuclein (αSyn) in dopaminergic neurons is regarded as a pivotal mechanism in PD pathogenesis. The accumulation of αSyn accelerates the aggregation of αSyn in dopaminergic neurons, and αSyn aggregates are associated with neuronal degeneration [5]. αSyn is a major component of Lewy bodies (LB) and Lewy neurites (LN). Most αSyn aggregating in vitro in LB and LN was found in the cytoplasm; however, some αSyn was localized in the nucleus [6,7,8,9]. In previous studies, nuclear αSyn manifested toxic effects on cells [7]. However, the mechanism of cell death induced by nuclear αSyn remains elusive. Our previous study showed that αSyn overexpression decreased nucleolin (NCL) levels in the nuclei of primary mouse embryonic fibroblasts (MEF), and enhanced oxidative stress-induced cytotoxicity and disturbance in ribosomal RNA (rRNA) processing in MEF [10]. Herein, we investigated the localization of αSyn accumulation in MEF that interfered with rRNA processing and promoted cytotoxicity; we also analyzed how the regulation of NCL levels could alter αSyn accumulation in MEF.

## 2. Results

### 2.1. Nuclear αSyn Increases Cytotoxicity Via Oxidative Stress

Because we desired to separate the pathological effect of the localization αSyn, we tagged the nucleotide sequence of nuclear export signal or nuclear localization signal on the αSyn. We transfected MEF with Flag-tagged wild-type αSyn (WT), αSyn containing nuclear export signal (NES), αSyn connected with nuclear localization signal (NLS), or vector plasmid DNA. These plasmid DNAs of vector, αSyn WT, NES, and NLS were persistently expressed for 48 h in MEF (Table 1). We confirmed that NLS was highly expressed in the nuclei of MEF, but a small amount of NLS was observed in the cytosol as well. NES was mostly found in the cytosol of MEF. However, WT was present in both the subcellular locations, and the expression of WT in the nuclei of MEF exhibited more intense immunostaining levels (Figure 1A). To verify the cytotoxic effects of the subcellular localization of αSyn, we measured the levels of cytotoxicity using lactate dehydrogenase (LDH) activity in culture media, as well as the levels of hydrogen peroxide in cells. We found that NLS showed the highest cytotoxicity via the generation of reactive oxygen species (ROS), which was represented by approximately 0.4 μM H_2_O_2_ levels (Figure 1B,C). The levels of H_2_O_2_ and the cytotoxicity induced by the expression of WT in MEF showed a slight increase, but the difference was not significant. It was envisaged that nuclear αSyn in MEF expressing WT might have induced a slight increase in H_2_O_2_ and cytotoxicity levels.

### 2.2. Nuclear αSyn Promotes Apoptosis Via Abnormal rRNA Processing

To validate the cytotoxicity induced by NLS expression in MEF, Western blot analysis was performed. As p53-mediated toxicity in PD patients or PD animal models was reported in the previous studies [11,12,13], we tested several apoptotic markers, such as p53, pro-caspase 3, and cleaved PARP. We found a significant elevation in apoptotic markers upon the ectopic expression of NLS in MEF. Along with the increase in apoptosis by nuclear αSyn, ribosomal protein S15 (RPS15) levels were increased upon NLS expression in MEF (Figure 2A,B). To test the abnormality of ribosomal biogenesis upon nuclear localization of αSyn, we quantified the levels of rRNA, including 28S and 18S rRNAs. We confirmed the successful transfection of Vector, WT, NES, and NLS, and the expression of Flag-tagged αSyn (Figure 3A). The levels of 28S rRNA of NLS showed a significant increase in MEF compared to the vector control; however, 18S rRNA levels were decreased upon the expression of NLS (Figure 3B). Taken together, the ratio of 28S rRNA to 18S rRNA was at the highest level as compared to that of the vector, WT, and NES (Figure 3C). However, the overexpression of a representative pathogenic mutant of αSyn, viz. alanine-30-prolin (A30P) and alanine-53-threonine (A53T), did not alter rRNA levels compared to the vector or GFP, as a control of protein overexpression (Appendix A). Several studies reported that the rRNA processing could be altered by the cellular stress, including oxidative stress [14,15]. NCL and p53 were known to be associated with the cellular stress [16], and the stress-induced abnormal conformations of nucleoli were reported in previous studies [17,18]. Stedman et al. reported that ribosome biogenesis dysfunction mediated p53-dependent apoptosis [19]. Our previous study demonstrated that nuclear NCL was decreased by the overexpression of αSyn [10]. Since the NCL is a major component of nucleoli, NCL immunostaining was used to validate whether the specific localization of αSyn can promote the disruption of the nucleolar structure. We observed that the segmentation of the NCL-positive nucleolar structure in MEF was increased by the expression of NLS (Figure 4A,B). These results suggested that nuclear localization of αSyn promotes abnormal ribosome biogenesis via NCL malfunction-mediated nucleolar stress, thereby leading to apoptosis.

### 2.3. The Control of NCL Levels Is Critical for the Clearance of αSyn

Our previous study has revealed that ectopic expression of NCL rescued synucleinopathy in MEF, rat primary cortical neurons, and mouse substantia nigra [10]. In this study, we tested the clearance of WT, NLS, and NES by knocking down NCL or ectopically expressing NCL in MEF. To validate the silencing of the *NCL* gene in MEF, we transfected MEF with short hairpin RNA (shRNA) for green fluorescent protein (shGFP) and shRNA for NCL (shNCL) for 24 h. We observed that NCL protein levels were significantly decreased and RPS15 levels were significantly increased following NCL repression (Figure 5A–C). In addition, the relative 28S/18S rRNA ratio was significantly increased upon the reduction in NCL levels (Figure 5D). Moreover, NCL downregulation resulted in a significant increase in the relative 28S/18S ratio compared with shGFP control in nuclear αSyn-expressing MEF (Appendix A). Surprisingly, a decrease in endogenous NCL significantly accelerated the accumulation of αSyn in the nucleus and cytoplasm (Figure 5E,F), and an increase in NCL resulted in a significant degradation of cellular localization-specific αSyn in MEF (Figure 6A,B). To test whether αSyn in MEF is released from cells to resolve the cytotoxicity induced due to αSyn accumulation, we assessed the culture media using our estimated sandwich-ELISA for aggregation formed-αSyn. However, no significant changes were observed among the various αSyn transfection culture media (Figure 6C). In conclusion, the control of NCL levels would be an effective therapeutic target for synucleinopathy, regardless of the subcellular localization of αSyn.

## 3. Discussion

Accumulation and aggregation of αSyn are critical for the progression of PD. However, the effects of the spatiotemporal presence of αSyn are not completely understood, and the toxic intermediates or the oligomeric state of αSyn responsible for the pathology of PD remain unclear. Most studies have emphasized the accumulation and aggregation of αSyn in cytoplasmic organelles, including endosomes, multivesicular bodies, autophagosomes, and lysosomes [20,21,22]. The cytoplasmic accumulation of αSyn increases the propagation of αSyn in dopaminergic neurons, as well as the elevation of neuroinflammation, thereby exacerbating PD progression [23,24]. In contrast, in this study, we revealed that the accumulation of αSyn in the cytosol and nucleus could result in distinct cytotoxicities in MEF. Several studies have reported that the presence of αSyn in the nucleus of cells may underlie αSyn-derived neurodegeneration [6,7,8,9]. We demonstrated that the nucleus-localized αSyn promoted similar cytotoxicity as described in previous studies [7,8,9], and was related to the abnormal rRNA processing and elevated ribosomal protein in this study (Figure 7). Thus, the neurotoxicity of nuclear αSyn might be mediated via ribosomal biogenesis, and the accumulated cytosolic αSyn would propagate the spread of alpha-synucleinopathy and neuroinflammation by the disruption of autophagy-lysosomal degradation pathway. On the other hand, αSyn-mediated neurodegeneration could be synergistically aggravated by both nuclear and cytosolic αSyn because the localization of WT αSyn has been observed in both the nucleus and cytoplasm.

The ectopic expression of NCL, which was regarded as a potential drug target for αSyn pathology in our previous study, alleviated αSyn-mediated ribosomal biogenesis and autophagy-lysosomal degradation pathway [10]. Moreover, the degradation of αSyn upon the ectopic NCL expression in MEF occurred regardless of the αSyn intracellular location (Figure 5). Several previous studies have demonstrated that NCL plays a critical role in rRNA processing [25,26,27]. Rotenone, a causative drug of PD, was found to decrease nuclear NCL levels in a previous study [28], and our previous report has revealed that the overexpression of αSyn decreased NCL levels in the nucleus [10]. Interestingly, an interaction between αSyn and NCL has been demonstrated in a previous study [29]. Hence, extensive interactions between αSyn and NCL in the nucleus might disrupt the formation of the nucleolus, and exacerbate the conformational and functional abnormalities of NCL. Moreover, the binding of an RNA, derived from a hexanucleotide repeat expansion (HRE), (GGGGCC)_n_, in the noncoding region of *C9orf72*, with NCL, could be a culprit behind amyotrophic lateral sclerosis and frontotemporal dementia via the aggravation of nucleolar stress [30,31]. Taken together, rRNA processing could be disrupted upon NCL dysfunction via interaction with αSyn in the nucleus.

Several studies have reported the involvement of ribosomal biogenesis in PD. Specifically, leucine-rich repeat kinase 2 (*LRRK2*), a causative gene of PD, has been reported as a gatekeeper of global mRNA translation by direct phosphorylation of ribosomal protein 15 [32,33]. DJ-1, another PD-related gene, was found to be localized in the stress granules and to interact with specific mRNA in dopaminergic neurons in a previous study [34]. Interestingly, another study has shown that p53 mRNA tended to be released from stress granules, and initiated CAP-independent translation via association with polyribosomes [35]. p53 activation has also been reported to be a key regulator of ribosomal biogenesis during erythroid differentiation [36]. Our previous study has revealed that p53 phosphorylation by LRRK2 is crucial for dopaminergic neural degeneration, and cellular senescence in dopaminergic neurons [37]. Hence, LRRK2 may be related to the juncture of ribosomal biogenesis stages via its crosstalk with other PD-associated genes or stressors. In addition, αSyn has been shown to be involved in ribosomal biogenesis in previous studies [38,39,40]. These observations might support our results that nuclear αSyn disrupts the nucleolar structure via malfunction of NCL, and enhances abnormal rRNA processing. Although we have validated the cytotoxicity by the impairment of rRNA processing, we should now investigate the distinct degeneration pathways of dopaminergic neurons via nuclear and cytosolic αSyn in a brain-like landscape, or in certain animal brains.

## 4. Materials and Methods

### 4.1. Cell Culture and Transfection

Although most primary cells from animals exhibited low transfection efficiency, MEF is known as a good cell model for the effect of ectopic gene expression on the primary cell. The effect of human αSyn overexpression and accumulation on some cells can be appropriate in MEF because the endogenous murine αSyn barely forms an aggregate itself, compared to the human αSyn. Hence, we used MEF for the cell model of this study. MEF were cultured using a growth medium composed of DMEM/F12 (LM 002-04; Welgene, Gyeongsan-si, Republic of Korea), 10% fetal bovine serum (BFS-1000; T&I, Chuncheon-si, Republic of Korea), and 1% antibiotic–antimycotic (100X) (15240062; Gibco, Carlsbad, CA, USA) in an incubator with 5% CO_2_ at 37 °C. For transfection experiments, 2 × 10^5^ MEF were seeded in a 12-well plate (30012; SPL, Pocheon-si, Republic of Korea). The next day, LipoD293 (SL100668; SignaGen Laboratories, Frederick, MD, USA) was used for the transfection of DNA plasmid (1.5 μg), including the vector pCDNA3.1, wild-type human Flag-tagged αSyn (WT), Flag-tagged wild type αSyn conjugated with with nuclear export signal (NES; 5′-CTGCAGCTGCCCCCCCTGGAGCGCCTGACCCTGGA-3′), and Flag-tagged wild type αSyn conjugated with nuclear localization signal (NLS; 5′-GATCCAAAAAAGAAGAGAAAGGTAGATCCAAAAAAGAAGAGAAAGGTA-3′). Following the transfection of various αSyn, we also sequentially transfected shGFP (1 μg) as a control for NCL silencing and shNCL (1 μg) to silence the *NCL* gene, or GFP (0.4 μg) and GFP-tagged NCL (0.8 μg) to test the degradation of αSyn [41]. Transfection was performed according to the manufacturer’s instructions.

### 4.2. Measurement of Cytotoxicity

The culture media of MEF transfected with the indicated plasmid DNA were collected to measure lactate dehydrogenase (LDH) activity, immediately. After 48 h of transfection, the harvest of cell lysates and the collection of culture media was simultaneously performed. The measurement procedure followed the guidance of the LDH Cytotoxicity Detection Kit (MK401; Takara Shuzo Co., Shiga, Japan). We set the treatment with 1% Triton X-100 for an hour in the growth medium as a control for 100% cytotoxicity and used the unused growth medium as a control for 0% cytotoxicity.

### 4.3. Analysis of Hydrogen Peroxide (H_2_O_2_) Level

Cells were washed with ice-cold Dulbecco’s Phosphate Buffered Saline (DPBS, LB 001-02; Welgene) twice and harvested with 60 μL of 1% Triton X-100 in DPBS. Lysates were centrifuged at 250 × g for 5 min at 4 °C. The supernatants and H_2_O_2_ standards were subjected to Amplex Red Hydrogen Peroxide/Peroxidase Assay Kit (A22188; Thermo Fischer Scientific, Waltham, MA, USA); the manufacturer’s instructions were followed.

### 4.4. Immunofluorescence Staining

A total of 2 × 10^4^ MEF were seeded on a poly-D-lysine (P7280; Sigma Aldrich, St. Louis, MO, USA)-coated 96-cellar well dark plate (655090; Greiner Bio-one, Kremsmünster, Austria), washed three times with ice-cold DPBS, and fixed with ice-cold 4% paraformaldehyde (PFA, 161-20141; FUJIFILM Wako Pure Chemical Corporation, Tokyo, Japan) for 15 min at room temperature (RT). After three washes with ice-cold DPBS, cells were permeabilized using ice-cold 0.1% Triton X-100 in DPBS for 5 min. We then washed the cells with ice-cold DPBS thrice, and blocked them using an antibody solution composed of 3% bovine serum albumin (BOVOSTAR, BSAS 0.1; Bovogen Biologicals, Keilor East VIC, Australia) and 1% normal goat serum (31873; Invitrogen, Carlsbad, CA, USA) in DPBS for an hour at RT. For the primary antibody reaction, DYKDDDDK Tag (D6W5B) Rabbit monoclonal antibody (1:1000, 14793; Cell Signal Technology, Beverly, MA, USA) and anti-C23 (NCL) mouse monoclonal (1:500, sc-55486; Santa Cruz Biotechnology, Dallas, TX, USA) were used in the antibody solution, and the mixture was allowed to react with the cells for 4 h at 4℃. After washing the cells thrice using ice-cold DPBS, the secondary antibodies, including Goat anti-Mouse IgG (H + L) Cross-Adsorbed Secondary Antibody, Alexa Fluor™ 488 (1:500, A-11001; Invitrogen) and Goat anti-Rabbit IgG (H + L) Cross-Adsorbed Secondary Antibody, Alexa Fluor™ 594 (1:500, A-11012; Invitrogen) were allowed to react with the cells for 2 h. The cells were then washed once with ice-cold DPBS, incubated with Hoechst33342 (62249; Thermo Fisher Scientific) in DPBS for 15 min, and then washed again with ice-cold DPBS twice. A mounting solution was added for subsequent microscopic analysis.

### 4.5. Western Blot Analysis

Cells were harvested with 1 × sample buffer diluted from Laemmli Sample Buffer (4X), Reducing (L1100-001, GenDEPOT Katy, TX, USA). Lysates were sonicated and boiled for 5 min for denaturation prior to their application to nitrocellulose membranes, following 10% sodium dodecyl sulfate-polyacrylamide gel electrophoresis. Western blotting was performed as described in our previous study. We used Luminata Crescendo Western HRP (WBLUR0500; Merck & Co., Inc., Kenilworth, NJ, USA) to develop immunoreactive signals on the nitrocellulose membrane, and the images were captured with a MicroChemi 4.2 camera (Shimadzu, Kyoto, Japan).

To detect the proteins of interest, we used the following antibodies: anti-RPS15 rabbit polyclonal (1:1000, CSB-PA020372LA01HU; CUSABIO, Houston, TX, USA), anti-C23 (NCL) mouse monoclonal (1:10,000, sc-55486; Santa Cruz Biotechnology), anti-Poly (ADP-ribose) polymerase (PARP) rabbit monoclonal (1:1000, 9542; Cell Signaling Technology), anti-Caspase3 rabbit monoclonal (1:1000: 9661; Cell Signaling Technology), anti-p53 mouse monoclonal (1:5000, sc-393031; Santa Cruz Biotechnology), anti-α-tubulin mouse monoclonal (1:5000, T5168; Sigma-Aldrich), anti-α-synuclein (clone 42) mouse monoclonal (1:10,000, 610786; BD Biosciences, San Jose, CA, USA), anti-GAPDH mouse monoclonal (1:3000, sc-32233; Santa Cruz Biotechnology), anti-β-actin mouse monoclonal (1:3000, sc-47778; Santa Cruz Biotechnology), goat peroxidase-conjugated AffiniPure anti-mouse IgG (H + L) (1:5000, #115-035-003; Jackson Immunoresearch Laboratories Inc., West Grove, PA, USA), and goat peroxidase-conjugated AffiniPure anti-rabbit IgG (H + L) (1:5000, #111-035-144; Jackson Immunoresearch Laboratories Inc.) antibodies.

### 4.6. mRNA Isolation and cDNA Synthesis

Cells were washed twice with DPBS warmed to 37 °C. We used the RNeasy Plus Mini Kit (74134; Qiagen, Germantown, MD, USA) for harvesting the cells and isolation of mRNA in MEF. TORscript cDNA synthesis kit (Enzynomic, Deajon, Republic of Korea) was used; the first strand cDNA was synthesized and amplified from 2 μg of total RNA. Mixed compounds made of 10X TOR script RT buffer 2 μL, TORscript Reverse Transcriptase (200 units/μL) 1μL, dNTP Mixture (2 mM) 2 μL, total RNA 2 μL, oligo dT primer 1 μL, RNase inhibitor (40 units/μL) 0.5 μL, and sterile water (RNase free) 11.5 μL, were incubated at 55 °C for 60 min. To inactivate the reaction, samples were immediately incubated at 95 °C for 5 min.

### 4.7. DNA Gel Electrophoresis

To validate the transfection DNA plasmid, cDNAs were amplified with the pair of Flag-tagged wild type αSyn primer. The PCR products were mixed with LoadingSTAR dye (Dyne Bio Inc., Seongnam, Republic of Korea), and dye-mixed samples were subjected to a 2% agarose gel made of HiQ Standard Agarose (A-0222-050, GenDEPOT). Images of the agarose gel were captured by G: BOX (Syngene, Bengaluru, India).

### 4.8. Quantitative PCR (qPCR)

The primers in Table 2 were used to measure target mouse rRNA. We used the Magnetic Induction Cycler (MIC, Bio Molecular Systems, Upper Coomera QLD, Australia) for carrying out qPCR. rRNA levels were estimated using the following equation: 2 ^ (-delta delta CT).

### 4.9. Sandwich Enzyme-Linked Immunosorbent Assay (ELISA)

To estimate the levels of αSyn after the transfection of αSyn DNA plasmid in MEF, we applied our αSyn aggregation-specific quantification ELISA tools using recombinant anti-alpha-synuclein aggregate antibody [MJFR-14-6-4-2]-conformation-specific (ab209538; Abcam, Cambridge, UK) and α-synuclein antibody (211) HRP (sc-12767 HRP; Santa Cruz Biotechnology). The culture medium was harvested and centrifuged at 4000 rpm for 10 min at 4 °C, and then ELISA was performed as described in our previous report (Figure 8).

### 4.10. Data Analysis

The Multi Gauge program (version 3.0) was used to measure the densities of the protein bands (FUJIFILM). Estimation and illustration of all datasets were performed using Prism 8 (GraphPad Software, San Diego, CA, USA). We have described the respective statistical analyses in each figure legend. All statistical tests that were used in this study are described in the figure legends.

## 5. Conclusions

Nucleus-localized αSyn is critical for cytotoxicity mediated by the disruption of rRNA processing or ribosomal protein biogenesis via the maintenance of NCL function for nucleolar conformation retention. Moreover, the elevation of NCL levels is essential for the degradation of accumulated αSyn.

## Figures and Tables

**Figure 1 ijms-24-02132-f001:**
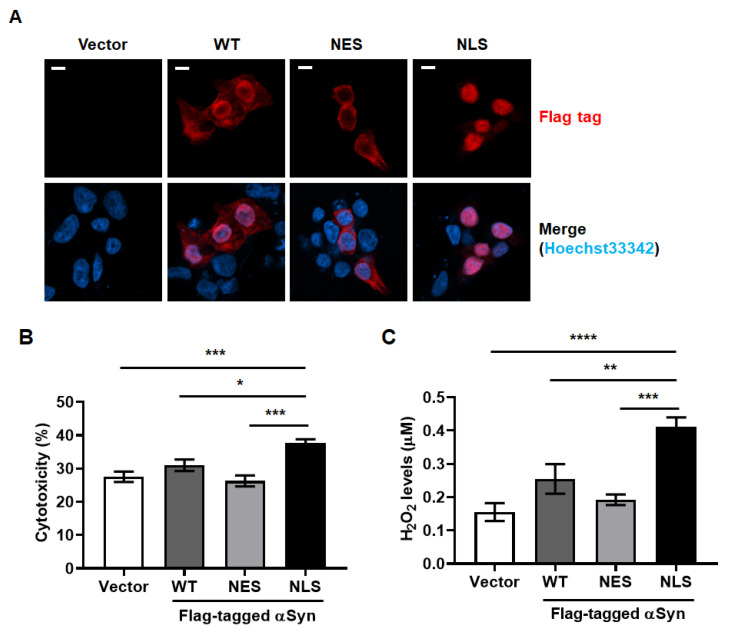
Nuclear localization of α-synuclein (αSyn) causes oxidative stress-induced cytotoxicity in mouse embryonic fibroblasts (MEF). (**A**) Distinct sub-cellular localization of αSyn was confirmed using immunofluorescent staining; red: Flag tag, Hoechst33342: nucleus. N = 3; white bar: 5 μm. (**B**) Lactate dehydrogenase (LDH) activity in MEF culture media was measured to represent cytotoxicity levels derived from the expression of indicated DNA, because the leakage of LDH from damaged cells represents cytotoxicity. Cytotoxicity (%) was calculated with the following: {(the absorbance of LDH activity of sample-the absorbance of negative control with unused normal growth media)/the absorbance of positive control with 1 % triton X-100 in growth media} X 100. WT: the Flag-tagged wild type αSyn, NES: Flag-tagged wild type αSyn conjugated with nuclear export signal, and NLS: Flag-tagged wild type αSyn conjugated with a nuclear localization signal. We used one-way analysis of variance (ANOVA) with Tukey’s post hoc test for statistical analysis. Data are represented as the mean ± SEM. N = 4; * *p* > 0.05 and *** *p* > 0.001. (**C**) The levels of hydrogen peroxide (H_2_O_2_) in MEF cytosol were estimated to analyze the cause of cytotoxicity derived from the distinct sub-cellular localization of αSyn in MEF. We used one-way ANOVA with Tukey’s post hoc test for statistical analysis. Data are represented as the mean ± SEM. N = 4; ** *p* > 0.01, *** *p* > 0.001, and **** *p* > 0.0001.

**Figure 2 ijms-24-02132-f002:**
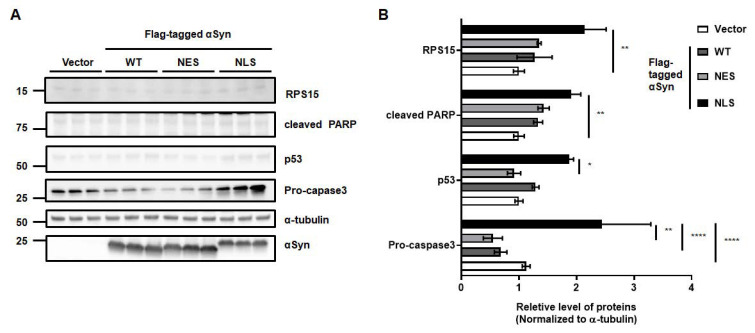
Nuclear localization of αSyn is responsible for apoptosis and ribosomal biogenesis in MEF. (**A**,**B**) The markers of apoptosis, such as pro-caspase3, p53, and cleaved Poly (ADP-ribose) polymerase (PARP), and ribosomal biogenesis marker, viz. 40S ribosome protein 15 (RPS15), were induced by the ectopic expression of nuclear αSyn. The densities of protein bands assessed using Western blotting were estimated and compared by the normalization with the density of β-tubulin band. We used one-way ANOVA with Bonferroni’s post hoc test. Data are represented as the mean ± SEM. N = 3; * *p* > 0.05, ** *p* > 0.01, and **** *p* > 0.0001.

**Figure 3 ijms-24-02132-f003:**
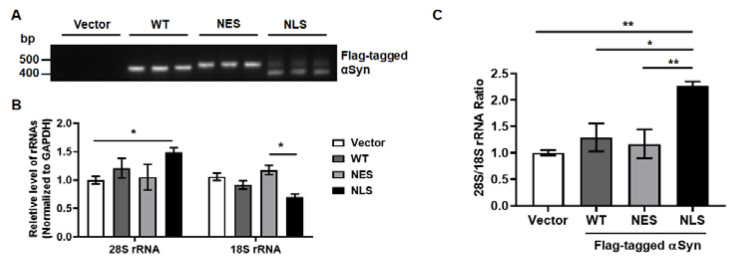
The levels of 28S and 18S rRNA are changed upon the nuclear localization of αSyn. (**A**) The validation of transfected DNA plasmid was confirmed with DNA gel electrophoresis. (**B**) 28S and 18S rRNA were analyzed with qPCR. The nuclear localization of αSyn promoted the induction of 28S rRNA and reduction in 18S rRNA in MEF. (**C**) The 28S/18S rRNA ratios were significantly increased by the nuclear localization of αSyn. We used one-way ANOVA with Tukey’s post hoc test. Data are represented as the mean ± SEM. N = 6; * *p* > 0.05 and ** *p* > 0.01.

**Figure 4 ijms-24-02132-f004:**
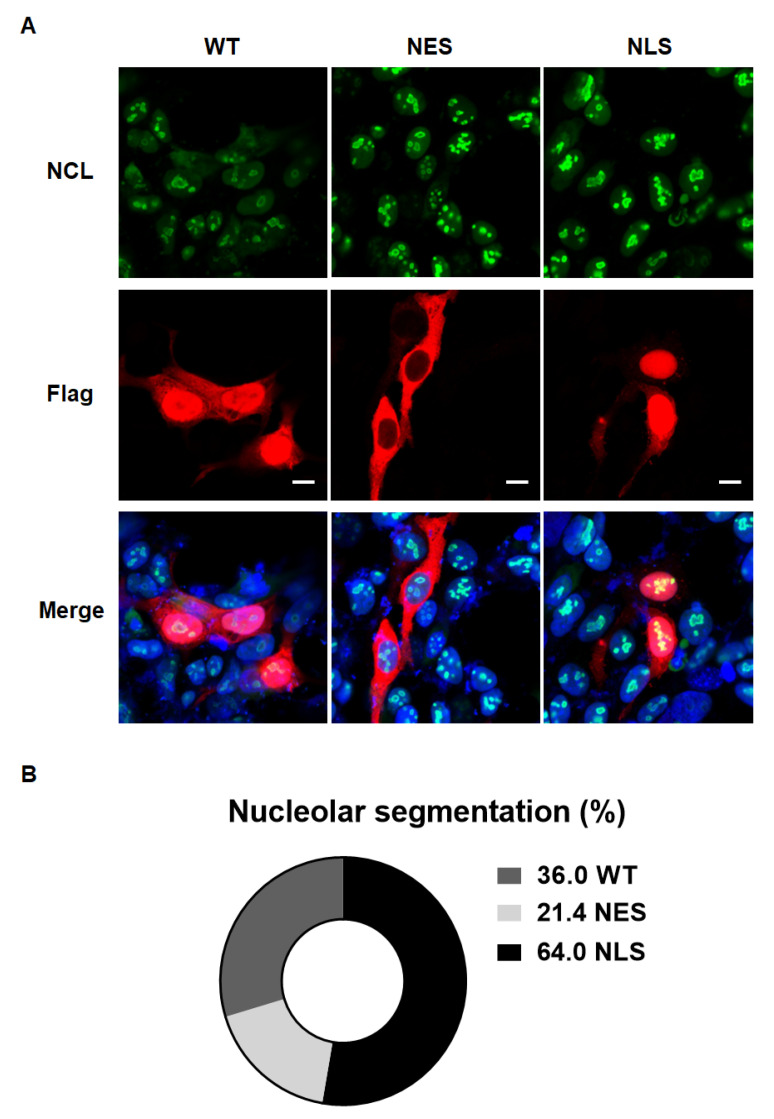
Nuclear localization of αSyn initiates the segmentation of nucleoli in MEF. (**A**) The morphology of nucleolus was analyzed using immunostaining of nucleolin (NCL) in MEF; green: endogenous NCL, red: Flag tag, blue: Hoechst33342. The number of cells with segmented NCL were counted by total transfected cells. (**B**) The transfection of MEF with nucleus-localized αSyn enhanced the segmentation of NCL-positive nucleoli. N = 4, number of cells = 15–18 per images (N); white bar: 5 μm.

**Figure 5 ijms-24-02132-f005:**
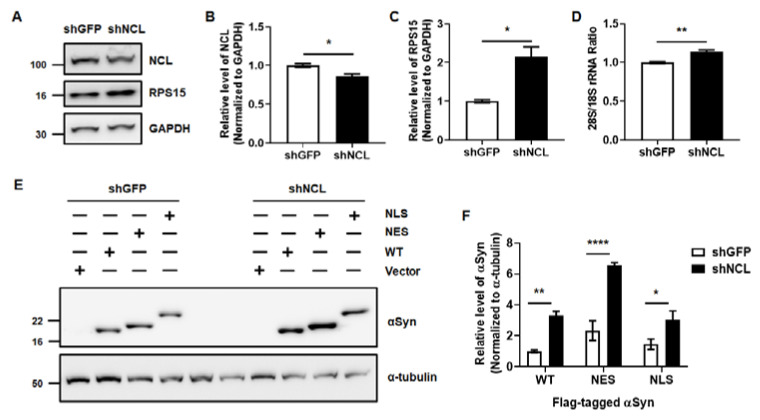
The reduction in NCL levels promotes αSyn accumulation. (**A**–**C**) The changes in NCL and RPS15 protein levels upon the transfection of shGFP and shNCL were analyzed. (**D**) 28S/18S rRNA ratio were analyzed using qPCR. Unpaired, two-tailed t-test was used for the comparison of results obtained from Western blot and qPCR analysis. (**E**,**F**) Following the transfection of various αSyn for 24 h, the sequential transfection of shGFP or shNCL for 24 h accelerated the accumulation of αSyn based on NCL levels in MEF. We used two-way ANOVA with Bonferroni’s post hoc test for analyzing the results obtained from Western blot analysis. Data are represented as the mean ± SEM. N = 3; * *p* > 0.05, ** *p* > 0.01, and **** *p* > 0.0001.

**Figure 6 ijms-24-02132-f006:**
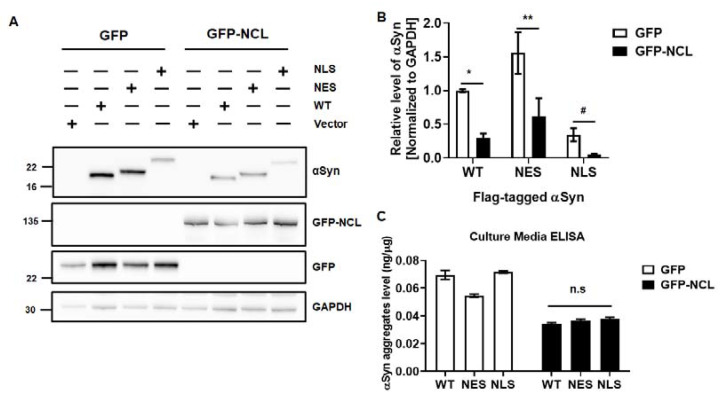
Ectopic expression of NCL induces the degradation of αSyn. (**A**,**B**) The sequential expression of ectopic GFP or GFP-NCL for 24 h, following the introduction of various αSyn for 24 h, led to the degradation of the accumulated αSyn in MEF. (**C**) The culture media used during various αSyn transfection were subjected to our αSyn sandwich-ELISA. We used two-way ANOVA with Bonferroni’s post hoc test for analyzing the results obtained from Western blot analysis. Data are represented as the mean ± SEM. N = 3; * *p* > 0.05, ** *p* > 0.01, and # *p* > 0.05 when using unpaired, two-tailed t-test, but n.s. when using two-way ANOVA. n.s.: not significant.

**Figure 7 ijms-24-02132-f007:**
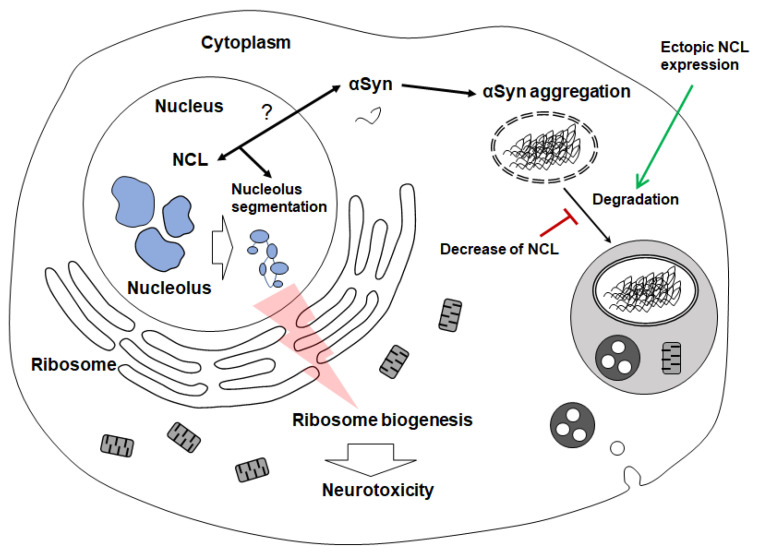
The graphical summary. Unlike most αSyn that accumulates in the cytoplasm, some αSyn that migrates to the nucleus might interact with NCL, resulting in the segmentation of nucleolus. Therefore, the disruption of nucleolar conformation could affect ribosome biosynthesis, thereby mediating neurotoxicity. The degradation of αSyn would be accelerated by the ectopic expression of NCL and repressed by the decrease in NCL. Question mark indicates the interaction between NCL and αSyn in the nucleus.

**Figure 8 ijms-24-02132-f008:**
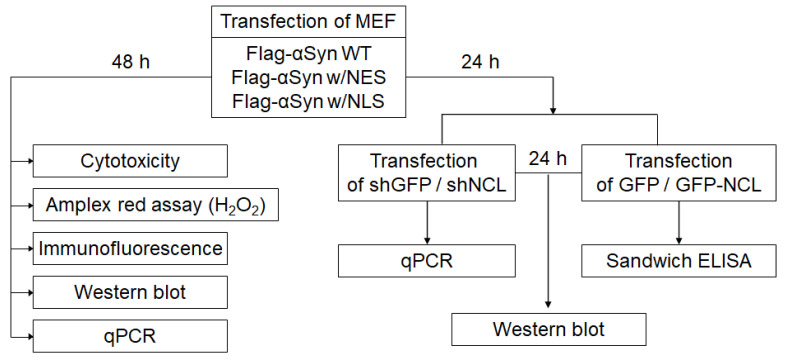
The methodological scheme. The experimental approaches that we applied in this study can be illustrated as the above workflow.

**Table 1 ijms-24-02132-t001:** The list of DNA plasmids.

DNA Plasmid	Origin	Abbreviation
pcDNA 3.1	-	Vector
Flag-tagged wild-type αSyn	Human	WT
Flag-tagged wild-type αSyn conjugated with nuclear export signal	Human	NES
Flag-tagged wild-type αSyn conjugated with nuclear localization signal	Human	NLS
Short hairpin RNA for green fluorescent protein	-	shGFP
Short hairpin RNA for nucleolin	-	shNCL
Green fluorescent protein	-	GFP
Nucleolin conjugated with green fluorescent protein	Human	GFP-NCL

**Table 2 ijms-24-02132-t002:** The sequence of primers for the Flag-tagged αSyn and mouse genes.

Genes		Sequence (5′–3′)	Annealing Temp (°C)
Flag tag	Forward	TACAAGGATGACGATGACAAGCTT	54
Human αSyn	Reverse	GGCTTCAGGTTCGTAGTCTTG	
28S rRNA	Forward	GGAAACTCTGGTGGAGGTCC	
28S rRNA	Reverse	CCTTAGCGGATTCCGACTTC	
18S rRNA	Forward	CTTAGAGGGACAAGTGGCG	
18S rRNA	Reverse	ACGCTGAGCCAGTCAGTGTA	
GAPDH	Forward	ACACTGAGGACCAGGTTGTC	
GAPDH	Reverse	TCCACCACCCTGTTGCTGTAG	

## Data Availability

The datasets generated and/or analyzed during the current study are available from the corresponding author upon reasonable request.

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
