# Peer review of "Nuclear α-Synuclein-Derived Cytotoxic Effect via Altered Ribosomal RNA Processing in Primary Mouse Embryonic Fibroblasts"

_ijms, 2023, doi:10.3390/ijms24032132_

Round 1

Reviewer 1 Report

My suggestions:

1. In the Methods section, I would add a figure on experimental workflow.

2. Authors may discuss NLS and NES signals in a little bit more in detail.  Either in the introduction or in the methods or in the results section.

3. P53 was increased in the case of NLS-expressed cytotoxicity. Is it possible that p53 was related to Parkinson-related toxicity? Is it possible that cancers and PD have a common pathway?

4. In the discussion I would add a pathway figure, on how nucleus-localized αSyn could play a role in neurotoxicity. 

5. In the methods section, for the qPCR I would add the table into tables.

Author Response

Dear reviewer

I sincerely appreciate your comment and kind pointing. We followed your suggestions and completed corrections and additions to the main manuscript. The revised parts were written with blue letter.

----------------------------------------------------------------------------------------------------------------

  1. In the Methods section, I would add a figure on experimental workflow.

- As you pointed, we added a scheme and legend for figure 8.

  1. Authors may discuss NLS and NES signals in a little bit more in detail. Either in the introduction or in the methods or in the results section.

- We added the DNA sequence of NLS and NES that we used in this study.

  1. P53 was increased in the case of NLS-expressed cytotoxicity. Is it possible that p53 was related to Parkinson-related toxicity? Is it possible that cancers and PD have a common pathway?

- Yes, it is. So, we revised the description at line 86-88 and added some references.  

  1. In the discussion I would add a pathway figure, on how nucleus-localized αSyn could play a role in neurotoxicity. 

- As you suggested, we added a graphical summary and legend for the brief conclusion of this study.  

  1. In the methods section, for the qPCR I would add the table into tables.

- As you mentioned, we added the table for qPCR primers.

Reviewer 2 Report

Review of a manuscript “Nuclear α-synuclein-derived cytotoxic effect via altered ribosomal RNA processing in primary mouse embryonic fibroblasts” by Dong Hwan Ho and coauthors submitted to IJMS

Parkinson’s disease (PD) is a prevalent severe neurodegenerative disorder for which there is no treatment modifying the course of the disease and no biomarkers for early diagnosis. One of the hallmark of the disease is the accumulation of misfolded and aggregated α-synuclein which may form toxic aggregate and larger protein inclusions. The authors presented new results on the localization and accumulation of α-synuclein in cells that interfered with rRNA processing and stimulated cytotoxicity. The manuscript also contains the analysis of the regulation of nucleolin levels and its effect on α-synuclein accumulation in mouse embryonic fibroblasts. This is an important field of biomedical study and these new results will be interesting for the readers of the journal.

The following corrections and additions should be made.

Abstract

Line 14: “α-Synuclein (αSyn) is a marker of Parkinson’s disease (PD).” Currently the results about the use of α-synuclein as Parkinson’s disease marker are controversial. So, it is better to replace this sentence by something like this one: “α-Synuclein (αSyn) is an important player in Parkinson’s disease (PD) pathogenesis.”

Lines 15-16: ”Most of the unfolded αSyn accumulates in the cytoplasm, but some αSyn has been found in the nuclei of neurons. To assess the effect of nuclear αSyn,…”. This text sounds confusing. The authors should clarify whether they mean only unfolded αSyn (as mentioned at the beginning of the first sentence) or total αSyn in the nuclei mentioned later.

Introduction

Lines 31-35: The authors refer to two PD reviews published a long time ago (1. Sawada, H., et al., 2004 and 2. Nagley, P., et al., 2010) and not covering all aspects of recent PD studies. They could replace them with two more recent comprehensive publications:

1 Oxidative stress and regulated cell death in Parkinson's disease. Ageing Res Rev. 2021 May; 67:101263. doi: 10.1016/j.arr.2021.101263.

2 Biomarkers in Parkinson’s Disease. Chapter in a book Peplow P.V., Martinez B., Gennarelli T.A. (eds) Neurodegenerative Diseases Biomarkers. 2022. Neuromethods, vol 173. pp 155-180. Humana, New York, NY. https://link.springer.com/protocol/10.1007/978-1-0716-1712-0_7

Line 40: “αSyn aggregates deteriorate the neuronal degeneration” what the authors mean?  May be they want to say that “:αSyn aggregates enhance the neuronal degeneration”?

Line 42: ”In previous studies, nuclear αSyn has shown toxic effects on cells [5].” This sentence should be corrected as follows:” In previous studies, nuclear αSyn manifested toxic effects on cells [5].”

Results

Lines 53: ”We transfected MEF with Flag-tagged wild-type αSyn (WT), nuclear export signal αSyn (NES), nuclear localization signal (NLS), or vector plasmid DNA.” The sense of this sentence is unclear. May be the authors mean: ”We transfected MEF with Flag-tagged wild-type αSyn (WT), αSyn containing nuclear export signal , αSyn connected with nuclear localization signal (NLS), or vector plasmid DNA.”?

Discussion

Lines 182-183: ”Several studies have reported that the presence of αSyn in the nucleus of cells may underlie αSyn-derived neurodegeneration [4-7]. Thus, the neurotoxicity of nuclear αSyn might be mediated via ribosomal biogenesis…”

The authors should explain why they make conclusion that neurotoxicity of nuclear αSyn might be mediated via ribosomal biogenesis…. No association is seen in this sentence

Author Response

Dear reviewer

I sincerely appreciate your suggestion and kind correcting. We followed your pointing and completed corrections and additions to the main manuscript. The revised parts were written with red letter.

---------------------------------------------------------------------------------------------------------------

Abstract

Line 14: “α-Synuclein (αSyn) is a marker of Parkinson’s disease (PD).” Currently the results about the use of α-synuclein as Parkinson’s disease marker are controversial. So, it is better to replace this sentence by something like this one: “α-Synuclein (αSyn) is an important player in Parkinson’s disease (PD) pathogenesis.”

- We changed line 14 as you suggested.

Lines 15-16: “Most of the unfolded αSyn accumulates in the cytoplasm, but some αSyn has been found in the nuclei of neurons. To assess the effect of nuclear αSyn…”. This text sounds confusing. The authors should clarify whether they mean only unfolded αSyn (as mentioned at the beginning of the first sentence) or total αSyn in the nuclei mentioned later.

- We changed line 15-16 to avoid confusing expression; “The aggregation of αSyn is mainly formed in cytoplasm, whereas some αSyn accumulation also has been found in the nuclei of neuron.”

Introduction

Lines 31-35: The authors refer to two PD reviews published a long time ago (1. Sawada, H., et al., 2004 and 2. Nagley, P., et al., 2010) and not covering all aspects of recent PD studies. They could replace them with two more recent comprehensive publications:

1 Oxidative stress and regulated cell death in Parkinson's disease. Ageing Res Rev. 2021 May; 67:101263. doi: 10.1016/j.arr.2021.101263.

2 Biomarkers in Parkinson’s Disease. Chapter in a book Peplow P.V., Martinez B., Gennarelli T.A. (eds) Neurodegenerative Diseases Biomarkers. 2022. Neuromethods, vol 173. pp 155-180. Humana, New York, NY. https://link.springer.com/protocol/10.1007/978-1-0716-1712-0_7

- We added two references that you suggested.

Line 40: “αSyn aggregates deteriorate the neuronal degeneration” what the authors mean?  Maybe they want to say that “αSyn aggregates enhance the neuronal degeneration”?

- We changed line 40; “αSyn aggregates is associated with the neuronal degeneration”

Line 42: “In previous studies, nuclear αSyn has shown toxic effects on cells [5].” This sentence should be corrected as follows:” In previous studies, nuclear αSyn manifested toxic effects on cells [5].”

- We changed line 42 as you pointed.

Results

Lines 53: “We transfected MEF with Flag-tagged wild-type αSyn (WT), nuclear export signal αSyn (NES), nuclear localization signal (NLS), or vector plasmid DNA.” The sense of this sentence is unclear. May be the authors mean: “We transfected MEF with Flag-tagged wild-type αSyn (WT), αSyn containing nuclear export signal , αSyn connected with nuclear localization signal (NLS), or vector plasmid DNA.”?

- We changed line 53 as you suggested.

Discussion

Lines 182-183: “Several studies have reported that the presence of αSyn in the nucleus of cells may underlie αSyn-derived neurodegeneration [4-7]. Thus, the neurotoxicity of nuclear αSyn might be mediated via ribosomal biogenesis…”

The authors should explain why they make conclusion that neurotoxicity of nuclear αSyn might be mediated via ribosomal biogenesis…. No association is seen in this sentence

- We added the description of results in this study to make sense. “By the way, we demonstrated that the nucleus-localized αSyn promoted similar cytotoxicity as described in previous studies and was related to the abnormal rRNA processing and elevated ribosomal protein in this study (Figure 7).”

Round 2

Reviewer 1 Report

The manuscript is acceptable. 

Author Response

We corrected that potins. And we also had a editing service by Editage. So we uploaded the certification.   
